# Acute effects of kinesiology tape tension on soleus muscle h-reflex modulations during lying and standing postures

**Yung-Sheng Chen**[1], **Wei-Chin Tseng**[1], **Che-Hsiu Chen**[2], **Pedro Bezerra**[3,4], **Xin Ye**[5,6]*

**1** Department of Exercise and Health Sciences, University of Taipei, Taipei, Taiwan, **2** Department of Sport Performance, National Taiwan University of Sports, Taichung, Taiwan, **3** Escola Superior de Desporto e Lazer, Instituto Politécnico de Viana do Castelo, Melgaço, Portugal, **4** The Research Centre in Sports Sciences, Health Sciences and Human Development, Vila Real, Portugal, **5** Department of Health, Exercise Science, and Recreation Management, The University of Mississippi, Oxford, Mississippi, United States of America, **6** Department of Rehabilitation Sciences, University of Hartford, Hartford, Connecticut, United States of America

* xye@hartford.edu

**Data Availability Statement:** All relevant data are within the manuscript and its Supporting Information files.

## Abstract

Kinesiology tape (KT) has been widely used in the areas of sports and rehabilitation. However, there is no gold standard for the tape tension used during a KT application. The purpose of this study was to examine the effects of KT application with different tension intensities on soleus muscle Hoffmann-reflex (H-reflex) modulation during lying and standing postures. Fifteen healthy university students were tested with 3 tape tension intensities during separate visits with a randomized sequence: tape-on no tension (0KT), moderate (about 50% of the maximal tape tension: (ModKT), and maximal tape tension (MaxKT). During each experimental visit, the H-reflex measurements on the soleus muscle were taken before, during, and after the KT application for both lying and standing postures. The H-wave and M-wave recruitment curves were generated using surface electromyography (EMG). There was a main effect for posture ($p = 0.001$) for the maximal peak-to-peak amplitude of the H-wave and M-wave ($H_{max}/M_{max}$) ratio, showing the depressed $H_{max}/M_{max}$ ratio during standing, when compared to the lying posture. Even though the tension factor had a large effect ($\eta_p^2 = 0.165$), different tape tensions showed no significant differential effects for the $H_{max}/M_{max}$ ratio. The spinal motoneuron excitability was not altered, even during the maximal tension KT application on the soleus muscle. Thus, the tension used during a KT application should not be a concern in terms of modulating the sensorimotor activity ascribed to elastic taping during lying and standing postures.

## Introduction

Kinesiology taping (KT) has been widely used in the areas of sports and physical therapy. This technique can potentially alter proprioception and functional performance of the taped muscle groups [1–3]. However, the taping effect on the motor functions is controversial in the

**Funding:** This study was supported by a research grant MOST 107-2410-H-845-021 from the Ministry of Science and Technology, R.O.C. (Taiwan).

**Competing interests:** The authors have declared that no competing interests exist.

literature, leading to reconsideration of current recommendations in KT technique. For example, facilitation of spinal motor excitability was reported when KT was applied to the calf muscles [4, 5]. Conversely, absence of taping effect on the motor functions was observed when the motor excitability at the supraspinal [6] and spinal levels was examined [7]. The discrepancy of research findings may be due to various methods of KT application, such as the tape tension and the tape direction among different studies.

In general, therapeutic practitioners adhere a tip of the tape strip onto the base (e.g., origin or insertion) of the target muscle, and remove the paper backing. Afterwards, the tape strip is stretched to a specific length that might represent a desired tape tension, and then simultaneously taped to the end of the muscle. The tape direction allegedly influences the motoneuron excitability (e.g., facilitation: tape applied from muscle origin to insertion; inhibition: tape applied from muscle insertion to origin). However, this still remains controversial [7]. In addition, practitioners tend to use their own preferences to stretch the tape. This individual-based variation can impose different tensions to the KT, which may potentially cause changes in the neuronal excitability due to a tactile stimulus. Currently, there is a lack of standardization for the optimal intensity of the tape tension applicable to the KT technique among practitioners. Notably, the optimal KT tension to improve physiological and functional performance varied among studies, and the best technique for applying KT remains to be controversial. For example, Velasco-Roldán et al. [8] reported no differential effects on pressure pain threshold and lumbar mobility in chronic low back pain patients when KT tape was applied with different tensions [e.g., no tension vs. light tension (15%-25% of the maximal tension) vs. moderate tension (40% of the maximal tension)]. In contrast, the optimal benefits were observed during a sensorimotor synchronization test when light and moderate KT tape tensions were applied to the wrist muscles, compared to non-tension KT [1, 2]. In addition, the application of 30%-40% and 50%-75% of the maximal KT tape tension both improved walking ability in amateur soccer players with lateral ankle sprain [9].

Postural control is an important sensorimotor function involved in daily life. It refers to the fine sensorimotor performance in spatial adjustment to the body positions [10, 11]. Somatosensory inputs detect changes in body positions via peripheral sensory receptors such as the muscle spindles, Golgi tendon organs, and joint receptors [12]. Overall, the motoneuron excitability [indicated by Hoffmann-reflex (H-reflex)] plays an essential role stabilizing body sway [13, 14]. For example, it has been reported that the peak-to-peak amplitude of H-reflex is depressed during standing, when compared to those during lying and sitting postures [15]. The depression of the H-reflex response during standing is considered as a protective mechanism to avoid oversaturation of the motor drive for postural maintenance [16, 17]. It is however, unclear, whether applying KT on different postures could alter the motoneuron excitability due to the potential changed sensory feedback, especially during a high tension KT session.

Firth et al. [5] first reported that the augmentation of motoneuron excitability (increase in the amplitude of the soleus H-reflex) was only observed in healthy participants but not in patients with Achilles tendinopathy, when applying the KT with inhibitory taping direction to the calf muscle. However, the taping technique used in this study had three different tape tensions in a single strip (50%-75% of the maximal KT tension over the Achilles tendon, 15%-25% of the maximal KT tension over the musculotendinous junction, and no tension over the top end of the tape). Thus, this method could potentially limit the reproducibility of the study. Tremblay and Karam [6] reported that KT application with the 50% maximal tape tension and facilitatory taping direction did not change the corticospinal excitability of the ankle muscles when one single KT strip was applied to the ankle dorsiflexors and plantar flexors. At the spinal level, Magalhães et al. [7] recently reported that facilitatory or inhibitory techniques

of KT applications to the calf muscles has no significant impact on the motor excitability in physically active adults. Both groups [6, 7] suggested that KT applications have little influence on the neuromuscular system and the elastic tensions are not enough to alter the sensory feedback, despite taping directions. Interestingly, the effects of high tension intensity (more than 75% of the maximal tape tension) KT applications on the neuromuscular system have rarely been examined [18], and it is possible that the varied results from the previous studies [19–22] were due to the inconsistent tape tensions among different experiments. With the greater tension of the elastic tape applied along the muscle belly, the pulling force generated by the tape might create more passive compression potentially shortening the muscle and its intrafusal fibres (e.g., muscle spindles), which could inhibit the taped muscle's H-reflex excitability [20].

Therefore, the purpose of this study was to examine the effects of KT application with different tension intensities [applying the tape with no tension (0KT), moderate (about 50% of the maximal tape tension: ModKT), and maximal tape tension (MaxKT)] on the soleus H-reflex modulation at lying and standing postures. It was hypothesized that MaxKT would show depressed soleus H-reflex modulation responses than the 0KT and ModKT conditions. In addition, the tension factor may be a moderator potentially altering the H-reflex modulation at different postures.

## Materials and methods

### Experimental approach to the problem

This investigation used a within-subjects crossover design to examine the potential effects of different KT tensions on the H-reflex responses in the soleus muscle at the lying and standing postures. All assessments and procedures were conducted in an exercise performance laboratory. The participants first visited the laboratory for a familiarization. Specifically, this visit started with measuring the subjects' physical characteristics (e.g., standing height, body weight). Following these measurements, the subjects were them familiarized with the H-reflex test, which mainly included surface EMG electrodes placements as well as locating the optimal sites of the stimulation electrodes. Lastly, the subjects were asked to practise two postures (lying and standing), during which the experimental procedures were explained. The second, third and fourth visits were experimental visits where different taping conditions (0KT, ModKT, and MaxKT) were randomly sequenced with at least 48 hours between consecutive visits. The experimental condition sequence order was conducted via Research Randomizer (https://www.randomizer.org/). Before (Pre), during (Mid), and after (Post) each condition, dependent variables were measured and recorded for further data analyses. In this study, the dominate leg was used to assess the reflex responses. Based on a previous study conducted by Firth et al. [9], moderate KT application had a moderate effect in facilitating motoneuronal excitability in healthy people. Thus, we conducted the a priori power analysis using the G*Power software (G*Power version 3.1.9.4, Düsseldorf, Germany) [23], on the basis of the possible large effect size (f = 0.40) for the difference in three tape tensions, an $\alpha$ level of 0.05, and a power (1-$\beta$) of 0.80. It was shown that at least 12 participants were necessary (F-test ANOVA, repeated measures, within factors).

### Subjects

Twenty-one healthy male students from the University of Taipei were initially recruited to participate in this study. Three participants were excluded due to their age and physical conditions. Eighteen participants (mean ± standard deviation: age = 21.3 ± 1.0 years; standing height = 1.75 ± 0.06 m; body weight = 70 ± 7.1 kg) completed the experiments but three of them were excluded for data analysis due to the absence or irregularity of the H-reflex

responses during the experiment. The inclusion criteria of participations included: 1) regular exercise participation (aerobic exercise, resistance exercise, or recreational sports activities) at least 3 times a week with accumulation of 150-min per week, and 2) age between 20–30 years old. Exclusion criteria included any history of severe neuromuscular injury, current lower extremity injury, and neurological diseases. All participants signed an informed consent form and undertook a familiarization session prior to the experiment. Participants were required to refrain from any strenuous activity 24 hours before participation. Based on which foot the participants would kick a soccer ball, 12 participants were right foot-dominant, and 3 were left foot-dominant. This study was approved by the Human Ethics Committee of the university (UT-IRB-2017-049) and was conducted in accordance with the Declaration of Helsinki.

## Experimental Procedures

**Surface electromyography electrodes placements.** After a minimum of 72 hours following the familiarization, the subjects returned to the laboratory for one of the experimental testing sessions. Each experimental visit lasted about 90 minutes. The locations of surface electromyography (EMG) electrodes on the soleus muscle were the exact locations determined during the familiarization visit. With both knees extended and relaxed, the participants first lay comfortably on a massage table with a prone position and with their feet hanging over the edge of the bed. The location of EMG placement was determined in accordance to the SENIAM project [24], at 2/3 of the distance from the medial condyles of the femur to the medial malleolus of the tibia and central to medial-lateral direction of the soleus muscle border. To replicate electrode placement across visits, the location of the electrode was outlined with indelible ink. Lastly, a reference electrode was placed above the medial malleolus of the non-dominant leg. Adhesive tapes were also used to secure the EMG electrodes on the skin. Surface EMG of the soleus was recorded through a data acquisition system (MP160, Biopac Systems, CA, USA) by using surface electrodes (TSD150B, Biopac Systems, CA, USA). The EMG sensors are active electrodes contained two stainless steel disks of 11.4 mm diameter with a 20-mm interdisk distance. The EMG signal acquisition was processed with common mode rejection ratio of 95 dB, signal to noise ratio > 89 dB, and input impedance of 100 M$\Omega$. The EMG signals were amplified (gain = 1000) and filtered with high and low pass filters set at 15 Hz and 500 Hz, respectively. The filtered signals were then digitized at a sampling rate of 2500 Hz, and stored in a laboratory computer for subsequent analyses.

**Stimulating electrodes placements.** After the EMG electrodes placements, the locations of the stimulation electrodes were then determined, with the same lying position on the massage table during the EMG electrodes placements process. Specifically, a reusable rubber-based self-adhesive electrode (10 x 10 mm, FA 25, Gem-Stick, Australia) was placed on the popliteal fossa as the cathode, and another square-shaped reusable rubber-based self-adhesive electrode (50 x 50 mm, Life Care, Taiwan) was fixed over the patella as the anode. Using an electrical stimulator with 1000 μs square pulse duration (DS7A, Digitimer Ltd, Herfordshire, UK), a single electrical impulse was delivered to the posterior tibial nerve to elicit the soleus H-reflex and M-wave responses. The location of the cathode electrode to elicit optimal H-reflex response was carefully checked several times by the principle investigator. After localizing the optimal site, it was marked with a permanent marker and the cathode electrode was attached to the participant's leg.

Following the placements of EMG and stimulating electrodes, the Pre-taping H-reflex assessment was conducted by a research staff. During the lysing posture, the participants remained prone during data collection. During the standing posture, the participants were informed to maintain a bilateral stance with both arms resting by their sides in a comfortable

position. A black dot, placed 2 meters in front of the participant at eye level was used as a visual target. The postures were randomly sequenced at the beginning of each experimental visit, and the sequence was maintained during that specific visit. After a specific posture, the transition to the next posture took approximately 30 seconds, and the participants were asked to maintain the next posture for at least one minute before the H-reflex assessment.

**Kinesiology taping.** With the completion of the pre-taping H-reflex assessment, the kinesiology tape was applied by a qualified physical therapist who held kinesiology taping certificate level 1, following the recommendations from the Kenzo Kase's Kinesiology taping manual. Specifically, a Y-shape kinesiology tape (Nitto Kinesiology Tape, NKH-X, Japan) with inhibitory direction was attached to the target locations. The baseline of the KT strips length was determined from calcaneal tubercle to the lateral condyle of femur and the medial condyle of femur. With the participants lying prone, the first strip of KT was applied alongside the border of lateral side of the gastrocnemius and the peroneus longus muscles. Another strip of KT was then placed alongside the border of the medial gastrocnemius and the soleus muscles. When the tape tensions increased to the ModKT and MaxKT, the insertion of the KT strips were attached along the boards of the hamstring muscles. While applying the KT, the participants fully extended their dominant knee joint with the ankle joint in a full dorsiflexion position. Each taping procedure took about 3 minutes, followed by 1 minute of rest at the predetermined posture, and then the assessments of the mid-taping H-reflex. Lastly, after the mid-taping H-reflex assessments, the physical therapist carefully removed the tape from the muscle in a tender manner to minimize the possible stimuli of cutaneous sensory inputs, and then the post-taping H-reflex assessments were conducted in the same manner as the previous assessments.

The intensity of the tape was determined by the length-tension relationship of the kinesiology tape, and adjusted by a physical therapist who conducted all KT applications in all experimental trials. Additionally, the individual calf length was also taken into consideration when applying the KT onto the muscles. A pre-experiment was conducted to test the tension force of KT from zero stretch tension to the maximal stretch. Specifically, the stretch tension of the kinesiology tape was assessed by using a custom-made push-pull gage box (Success Push-Pull Gage, AEF-2, Japan). The baseline of tape length was set at 16 cm as a zero-stretch tension (0KT), and the tape stretch tension was measured with increment of stretch length every 0.5 cm until the maximal stretch intensity (the point where a strong resistance could be felt, thus passing this point could lead to the fracture of the tape). Each length was assessed 5 times and average tension value was reported (Fig 1). The physical therapist applied the 0KT, ModKT, and MaxKT conditions based on the percentage of length-tension ratio of the kinesiology tape and the length of the calf muscles. A research assistant was supported removal of the paper backing in order to carefully control the tape tension by the physical therapist.

### Measurements and data analyses

**H-reflex assessment.** During the H-reflex assessment, firstly the stimulation intensity was increased with 10 mA increments from the H-reflex threshold ($H_{threshold}$) until the plateau of the maximal peak-to-peak amplitude of the M-wave ($M_{max}$) was identified, to establish the H/M recruitment curve. Specifically, when the first plateau intensity of stimulation for $M_{max}$ was identified, we used 150% of this intensity to record the $M_{max}$. Secondly, the maximal peak-to-peak amplitude of H-reflex response ($H_{max}$) was then determined by using 2 mA increment from the stimulation intensity of the $H_{threshold}$ in the subsequent determination. Each stimulation intensity was applied four times with at least 10 seconds inter-stimulation interval [12]. The number of stimuli for determination of H/M recruitment curve varied from individual to

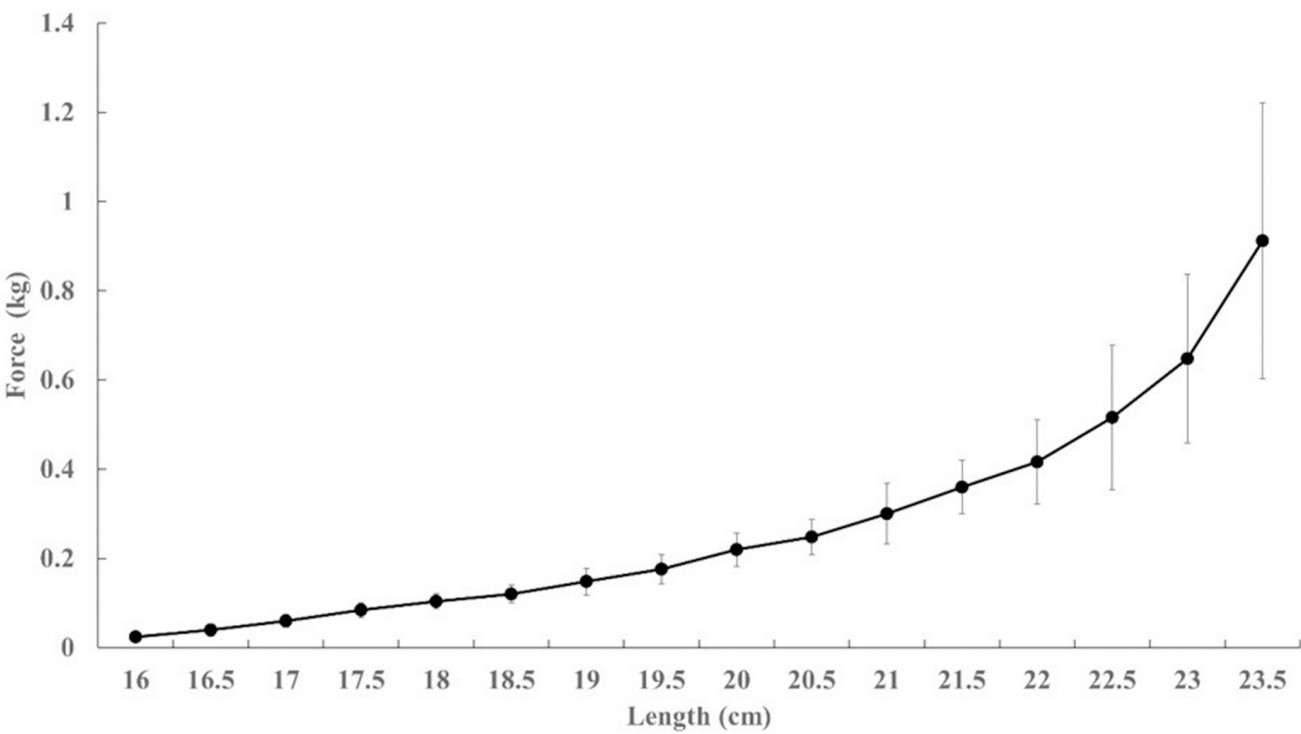

**Fig 1. Tape length and tension relationship of the kinesiology tape used in the current experiment.**

individual, with a range of 40–60 stimuli in each experimental condition. A data acquisition and analysis system (AcqKnowledge 5.0, Biopac, CA, USA) was used to synchronize the electrical stimulations and the EMG recording. The peak-to-peak amplitude of raw EMG was used to assess the size of reflex responses and then normalized as a percentage of the amplitude value corresponding to the muscle's $M_{max}$.

## Data analyses

The $H_{max}$ and $M_{max}$ were measured off-line for each experimental condition. The average value of H-reflex and M-wave variables in all successful trials of each posture and tension were processed by one researcher. The $H_{max}/M_{max}$ ratio was then calculated, which indicates the proportion of the entire spinal motoneuron pool activated by the Ia afferent inputs during the reflex response [12].

## Statistical analyses

Descriptive data of the measured variables were presented as mean ± SD. After the examination and confirmation of normal distribution of each variable with the Kolmogorov-Smirnov test, separate three-way repeated-measures analysis of variance tests (ANOVAs) (Tension [3] x Time [3] x Posture [2] were used to examine the dependent variables. When appropriate, the follow-up tests included Bonferroni-adjusted pairwise comparisons. The $\eta_p^2$ statistic is provided for all repeated measure comparisons, with values of 0.01, 0.06, and 0.14 corresponding to small, medium, and large effect sizes, respectively [25]. An alpha value of $p \leq 0.05$ was set for significant differences between the means. All statistical analyses were performed by SPSS® Statistics version 24.0 (IBM, Armonk, NY, USA).

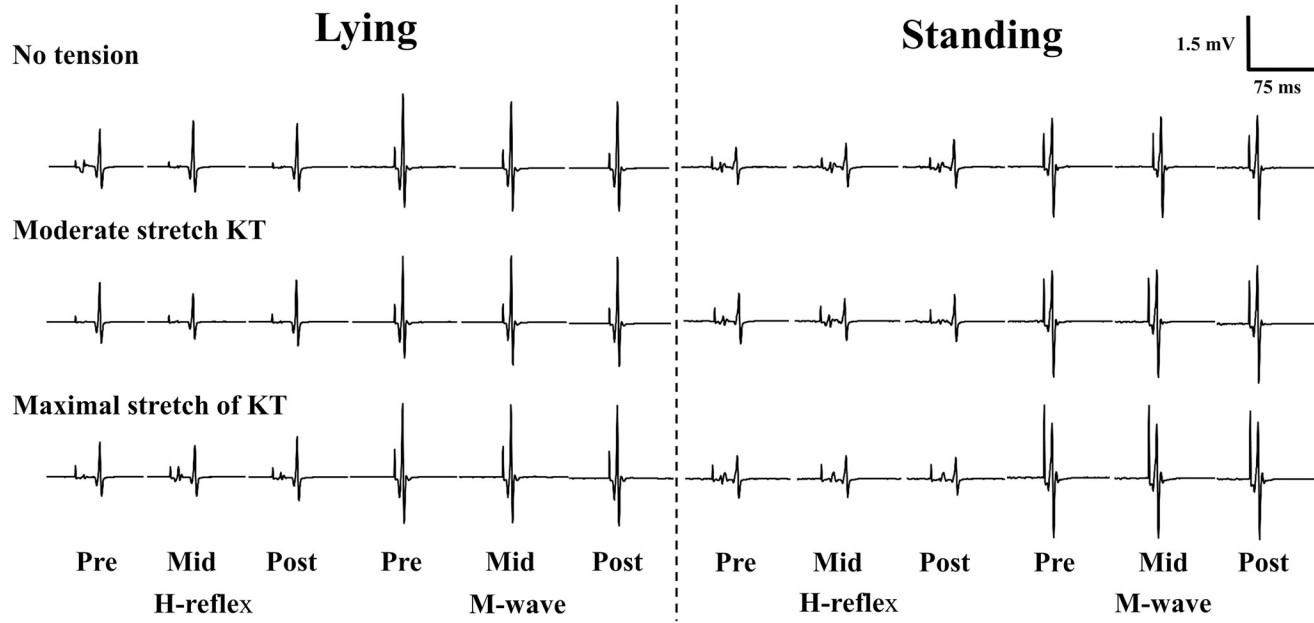

**Fig 2. The example of a participant's H-reflex and M-wave recoding tracings before (Pre), during (Mid), and after (Post) different KT intensities at the lying (A) and standing (B) postures.**

## Results

Mauchly's test of Sphericity was used to check the violation of the assumption of sphericity for all the repeated measures ANOVA tests. If violated, Greenhouse-Geisser corrected statistics were then used. Fig 2 shows examples of H-reflex and M-wave recording traces. Fig 3 shows an example of a participant's H-wave and M-wave recruitment curve before, during, and after different KT intensities with different postures. For both the $H_{threshold}$ and $H_{max}$, the three-way ANOVAs showed no significant 3-way or 2-way interactions, but there was a main effect for posture ($H_{threshold}$: p = 0.005; $H_{max}$: p = 0.001), indicating significantly greater $H_{threshold}$ and $H_{max}$ values during lying than those during the standing posture (Table 1). For the $M_{max}$, the three-way ANOVA showed no significant 3-way or 2-way interactions, but there were main effects for posture (p = 0.038), and tension (p = 0.033). The follow-up pairwise comparisons indicated significantly greater $M_{max}$ value during lying than those during the standing posture (Table 1). In addition, the $M_{max}$ value was significantly lower (posture and time merged mean: 2.602 vs. 2.831 mV, p = 0.046; Table 1) during 0KT than that during ModKT. The three-way ANOVA for $H_{max}/M_{max}$ ratio indicated no significant 3-way or 2-way interactions, but there was a main effect for posture (p = 0.001), indicating significantly greater $H_{max}/M_{max}$ ratio during lying than those during the standing posture. Table 1 summarizes the mean difference and 95% confidence interval of the pairwise comparisons among the three different tensions (posture and time merged) for all H-reflex and M-wave variables.

## Discussion

The purpose of this study was to examine the effects of KT application with the different tension intensities on soleus muscle spinal motoneuron pool excitability (H-reflex parameters) at lying and standing postures. The main findings of this study include: 1) the H-reflex parameters (raw H-reflex amplitude and $H_{max}/M_{max}$ ratio) were significantly lower during the standing posture, when compared to the lying posture, regardless the different tension intensities or

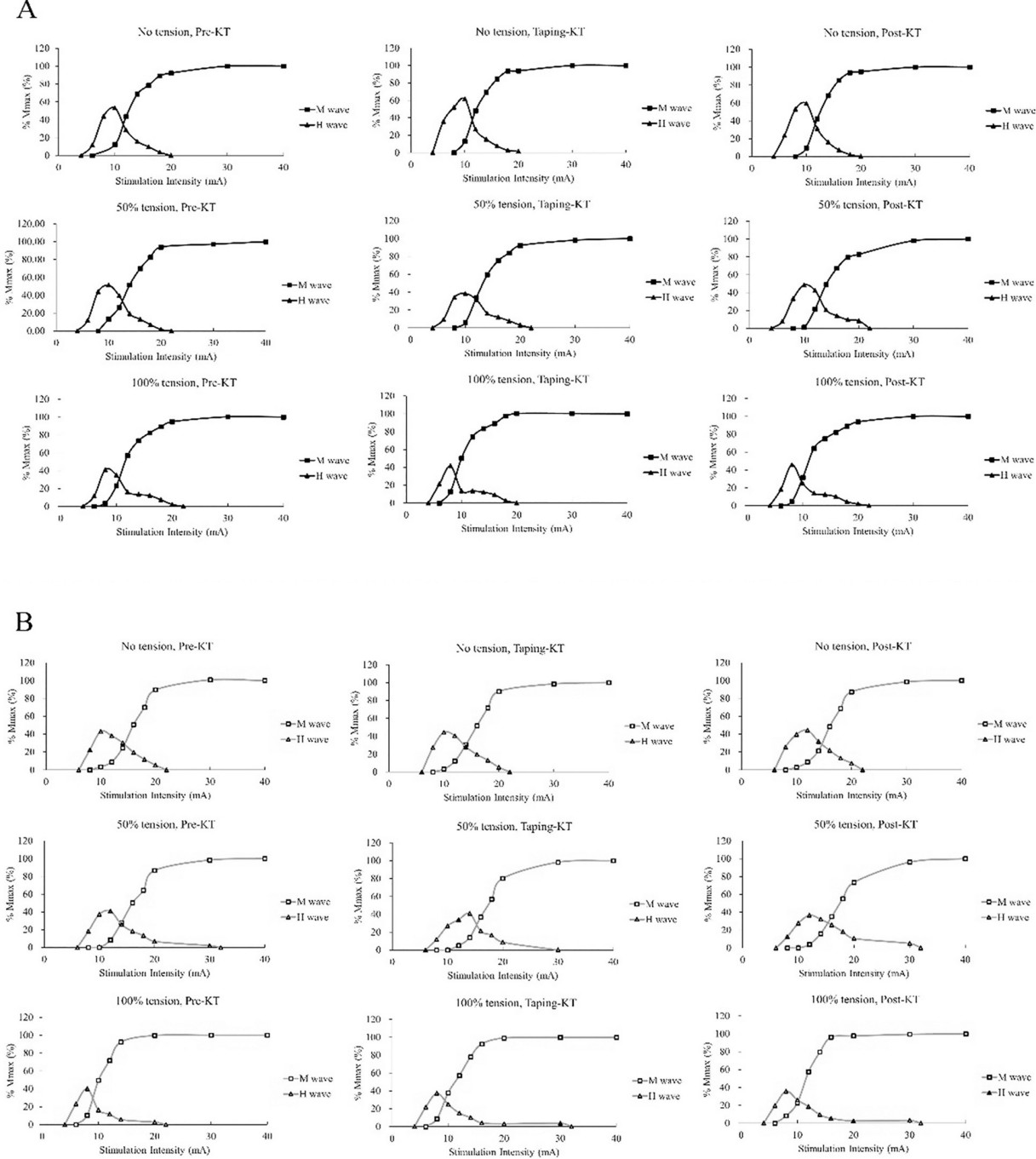

**Fig 3. The example of a participant's H-wave and M-wave recruitment curve before (Pre), during (Mid), and after (Post) different KT intensities at the lying (A) and standing (B) postures.**

measurement time points, and 2) the tape tension factor did not influence the H-reflex modulation.

Regarding the H-reflex modulation during lying and standing postures, our results were within the expectation: the raw H-reflex amplitude and $H_{max}/M_{max}$ ratio were both significantly lower during the standing posture, when compared to the lying posture, regardless the

**Table 1. Mean (SD) of all H-reflex and M-wave variables (merged across time) on each tension (0KT, ModKT, and MaxKT) for lying and standing postures. Mean difference (MD) along with the 95% confidence interval (95CI) of the pairwise comparisons for tensions (posture and time merged) were also presented**.

| | | 0KT | MD (95CI) 0KT vs. ModKT | ModKT | MD (95CI) ModKT vs. MaxKT | MaxKT | MD (95CI) 0KT vs. MaxKT |
|---|---|---|---|---|---|---|---|
| $H_{Threshold}$ (mV) | Lying | 0.91 (0.63) | -0.061 (-0.334, 0.212) | 0.99 (0.69) | 0.057 (-0.139, 0.254) | 0.84 (0.59) | -0.003 (-0.297, 0.291) |
| | Standing | 0.56 (0.23) | | 0.55 (0.31) | | 0.61 (0.36) | |
| $H_{Max}$ (mV) | Lying | 1.74 (0.77) | -0.040 (-0.248, 0.167) | 1.78 (0.82) | 0.079 (-0.107, 0.265) | 1.65 (0.79) | 0.038 (-0.132, 0.209) |
| | Standing | 1.09 (0.43) | | 1.13 (0.49) | | 1.10 (0.45) | |
| $M_{Max}$ (mV) | Lying | 2.81 (0.58) | -0.229 (-0.487, 0.030) | 2.94 (0.46) | 0.049 (-0.180, 0.279) | 2.98 (0.58) | -0.179 (-0.396, 0.037) |
| | Standing | 2.39 (0.61) | | 2.72 (0.55) | | 2.58 (0.52) | |
| $H_{Max}/M_{max}$ Ratio | Lying | 0.61 (0.21) | 0.033 (-0.029, 0.094) | 0.59 (0.24) | 0.014 (-0.050, 0.078) | 0.55 (0.24) | 0.047 (0.008, 0.086) |
| | Standing | 0.46 (0.16) | | 0.42 (0.16) | | 0.43 (0.17) | |

different tension intensities or measurement time points. It has been well-established that the spinal motoneuron pool excitability is depressed during the standing posture, when compared to other postures (e.g., lying and sitting) [15, 26]. Specifically, this phenomenon is likely influenced by presynaptic inhibitory mechanisms at the cortical [27] and/or the spinal levels [28]. Decrease in the H-reflex responses is thought as a major factor to avoid oversaturation of the descending efferent inputs to the target motoneurons for fine control of sensorimotor activities during upright standing. Furthermore, we also expected to see different KT tensions would alter the sensory feedback of the testing muscles, consequently change the neural behavior during different postural tasks. However, the intensity of KT stretch tension showed no effect on the soleus H-reflex modulation between lying and standing in the present study.

Previously, research studies have been conducted to examine the effects of KT tape tension on functional variables (e.g., strength, sensorimotor coordination, pressure pain threshold, mobility) in healthy adults and special populations [1, 8, 18]. The majority of these parameters were not altered by the tightness or tension of the KT application. Similarly, our results show no effect of KT tape tensions on the soleus H-reflex modulation. It was our intention to create different tape tension intensities on the participants' soleus muscle during the KT application. And we hypothesized that the highest tape tension intensity would decrease the H-reflex, due to the passive compression of the skin and connective tissue under the tape, generated by the elastic pulling force. However, the $H_{max}/M_{max}$ ratios were not statistically significantly different among different tensions. Acute changes in the H-reflex can be due to alterations of the spinal motoneurons under different conditions [29]. Thus, the current finding suggests that the KT application even with the highest tape tension did not alter the motoneuron excitability of the soleus muscle. A previous experiment [6] reported that a 50% maximal tape tension KT application did not change the corticospinal excitability of the ankle muscles. In addition, at the spinal level, Magalhães et al. [7] found KT applications to the calf muscles has no significant impact on the motor excitability in physically active adults, regardless of the tape direction. Both groups [6, 7] suggested that KT applications have little influence on the neuromuscular system and the elastic tensions are not enough to alter the sensory feedback. Our measurement test on tension of the kinesiology tape used in the current study indicated a maximum tension of 1kg, which might still be too low to induce any significant changes to the muscle, leading to the unchanged H-reflex modulation. It is, however, important to point out the large effect size ($\eta_p^2 = 0.165$) of the tension effect on the $H_{max}/M_{max}$ ratio (Table 1). Thus, theoretically, if the tape tension kept increasing (e.g., passing the KT tape's maximal stretch limitation) when conducting the KT application, the spinal motoneuron excitability could have been manipulated. However, with the current KT tape, the tension doesn't seem to provide meaningful benefits in terms of manipulating spinal motoneuron excitability.

Even though the current study had some interesting findings, this experiment did have some limitations. First, the participants recruited in the present study were physically-active college students. They were weekly involved recreational basketball training, amateur take-down training, running, cycling or resistance training. The background of their physical activity levels, along with their healthy statuses, make the current results not necessarily be readily applicable to other populations (e.g., injured athletes who are going through rehabilitation). Second, KT application applied at the prone position may not provide the same tape tension as that at the standing position. As mentioned, the potentially altered ankle position (due to tape tension) of the relaxed leg during the prone position could possibly induce changes in the H-reflex parameters, which may limit our interpretation of the results at this posture. Third, we only focused on examining the H-reflex parameters, rather than incorporating some functional tests (e.g., balance test) in the current study. A lack of functional outcome measures means that the findings are limited to the exploration of reflex excitability rather than any functional impact of tape.

In conclusion, the current experiment used different tape tensions during KT application on the soleus muscle at both standing and lying postures. The spinal motoneuron excitability was depressed at the standing, when compared to that during the lying posture. However, the changes in tape tension were not accompanied by the changes in spinal motoneuron excitability, even during the maximal tape tension KT application. Our findings provide important information for practitioners such as physical therapists and athletic trainers regarding the potential influence of KT tension intensity on the soleus muscle spinal motoneuron excitability: the tape tension does not seem to alter the spinal excitability.

## Supporting information

**S1 Data.**
(SAV)

**S2 Data.**
(SAV)

## Acknowledgments

The authors would like to thank all the subjects who took time out of their schedules to help with this project.

## Author Contributions

**Conceptualization:** Yung-Sheng Chen, Xin Ye.

**Data curation:** Yung-Sheng Chen, Wei-Chin Tseng, Pedro Bezerra, Xin Ye.

**Formal analysis:** Yung-Sheng Chen, Wei-Chin Tseng, Pedro Bezerra, Xin Ye.

**Funding acquisition:** Yung-Sheng Chen.

**Investigation:** Yung-Sheng Chen, Che-Hsiu Chen, Xin Ye.

**Methodology:** Yung-Sheng Chen, Wei-Chin Tseng, Che-Hsiu Chen, Pedro Bezerra, Xin Ye.

**Project administration:** Yung-Sheng Chen.

**Resources:** Wei-Chin Tseng, Che-Hsiu Chen, Pedro Bezerra, Xin Ye.

**Software:** Yung-Sheng Chen, Wei-Chin Tseng, Che-Hsiu Chen, Pedro Bezerra, Xin Ye.

**Supervision:** Xin Ye.

**Validation:** Che-Hsiu Chen, Pedro Bezerra, Xin Ye.

**Visualization:** Che-Hsiu Chen.

**Writing – original draft:** Yung-Sheng Chen, Xin Ye.

**Writing – review & editing:** Yung-Sheng Chen, Wei-Chin Tseng, Che-Hsiu Chen, Pedro Bezerra, Xin Ye.

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
