## [Decision Letter · Decision Letter 0]

1 Apr 2020

PONE-D-20-05949

Acute Effects of Kinesiology Tape Tension on Soleus Muscle H-reflex Modulations during Lying and Standing Postures

PLOS ONE

Dear Dr. Ye,

Thank you for submitting your manuscript to PLOS ONE. After careful consideration, we feel that it has merit but does not fully meet PLOS ONE’s publication criteria as it currently stands. Therefore, we invite you to submit a revised version of the manuscript that addresses the points raised during the review process.

Your manuscript has been reviewed by an expert in the field . I have also examined the paper myself since the second reviewer could not provide a report in due time, given the current crisis with COVID. As you will see below, the Reviewer has expressed several significant concerns. First, the Reviewer is right in pointing that both the introduction and discussion need to be revised to provide a more balanced account of the current literature regarding reported neurophysiological effects of KT applications. Second, the Reviewer points to several issues with the methodological approach (e.g., sample size justification, participants' characteristics, tape tension). Finally, the Reviewer has questions regarding the statistical approach (e.g., justification for using t-tests, reporting of effect size). I fully concur with the Reviewer's report. The paper needs substantial revisions, and you need to make sure that all concerns and issues are adequately addressed if you opt to submit a revised version.

We would appreciate receiving your revised manuscript by May 16 2020 11:59PM. To enhance the reproducibility of your results, we recommend that if applicable you deposit your laboratory protocols in protocols.io, where a protocol can be assigned its own identifier (DOI) such that it can be cited independently in the future. For instructions see: http://journals.plos.org/plosone/s/submission-guidelines#loc-laboratory-protocols

We look forward to receiving your revised manuscript.

Kind regards,

François Tremblay, PhD

Academic Editor

PLOS ONE

Journal Requirements:

2. Please include additional information regarding the recruitment of participants to this study. For example, where were they recruited from and what sampling method was used. In addition, please further describe the "a priori power analysis" performed as little detail is provided.

Reviewers' comments:

Reviewer's Responses to Questions

**Comments to the Author**

1. Is the manuscript technically sound, and do the data support the conclusions?

Reviewer #1: Partly

2. Has the statistical analysis been performed appropriately and rigorously? 

Reviewer #1: Yes

3. Have the authors made all data underlying the findings in their manuscript fully available?

Reviewer #1: Yes

4. Is the manuscript presented in an intelligible fashion and written in standard English?

Reviewer #1: No

5. Review Comments to the Author

Reviewer #1: INTRODUCTION

• Lines 54-55: I disagree with this information, as there are several studies demonstrating that, in fact, kinesiotaping does not alter the proprioceptive responses or has a minimally effect. I suggest that you present the current evidence and present the gaps that justify your study. For instance, there are at least two studies using H-reflex measurements within the KT context, showing that the tape does not influence the motoneuron excitability:

https://journals.humankinetics.com/view/journals/jsr/aop/article-10.1123-jsr.2018-0435/article-10.1123-jsr.2018-0435.xml

https://pubmed.ncbi.nlm.nih.gov/21079436/

• Lines 58-59: You should further discuss the issue of KT tension and how it is applied, because there are several studies showing wide variability. This raises the issue of a poor reliability regarding the tape tension. In addition, I recommend that you discuss the issue of tape direction, which supposedly influences the motoneuron excitability (by facilitation or inhibition). This is an important aspect that is still controversial.

METHOD

• Please explain in detail the familiarization procedure

• The sample size calculation has missing information. For instance, what was the statistical model adopted for the calculation? Also, what was the effect size considered? And what was the outcome used for the analysis?

• If the participants were physically active and adopted a regular physical exercise routine, how did you manage to discriminate between the effects of their daily routine and the effects of the elastic tape? What kind of exercise they performed? You should include this information as a characterization.

• The post taping h-reflex assessment was performed after the removal of the tape? In this case, how did you manage to control the taping effects in the skin (a possible cutaneous stimulation), from the assessment of the h-reflex?

• Regarding the tape tension: the procedure is not clear. For instance, how did you manage to control the tension-length ratio, in order to avoid the fracture point? Also, if you reached the maximum stretch intensity, how did you guarantee that this intensity did not compromised the elastic properties of the tape? Several studies adopted equations to control this aspect, by using a percentage of tape tension, in order to reproduce the original instructions of the method… please clarify.

• It is not clear what was the purpose of the VAS scale. I fail to see any justification to investigate the participant’s perception. Please clarify.

• Statistical analysis:

o Lines 243-245: this information is confusing. Why did you adopt t tests?

o Cohen’s d should be used in this case, but only the eta partial squared. Cohen’s d is used for bivariate comparisons, which is not the case. Please correct accordingly.

RESULTS

• Overall, the results section does not read well. I would recommend to be more objective within the textual presentation of the results, keeping the numbers to a minimum. Details of effect size, mean differences, and so on, should be presented in tables. This would improve the reading.

• Please include mean differences and confidence interval of 95% of comparisons.

• Table 1 is confusing because it presents the comparisons, but not the data.

DISCUSSION

• Lines 301-305: the main findings present a causal relationship, for instance, that posture plays a significant role influencing h-reflex parameters. However, in order to reach this conclusion a parallel design would have been more adequate, and not a repeated measure design with the same participants.

• Again, what was the purpose of the VAS? This result does not add to the study, as the authors might have included in order to show that “something happened” and the participants perceived the tension. However, this might bias the readers, as your focus was the motoneuron excitability, and not subject’s perception.

• Lines 351-352: this information is biased, as several studies investigated tape tension and tape direction.

• Lines 368-371: there are several studies that already investigated the KT effects within functional performance. This should be discussed in light of your findings.

6. PLOS authors have the option to publish the peer review history of their article (what does this mean?). If published, this will include your full peer review and any attached files.

Reviewer #1: No

---

## [Author Response · Author response to Decision Letter 0]

10 May 2020

Please see our "Responses to Reviewer" word document for detailed responses.

---

## [Decision Letter · Decision Letter 1]

21 May 2020

PONE-D-20-05949R1

Acute Effects of Kinesiology Tape Tension on Soleus Muscle H-reflex Modulations during Lying and Standing Postures

PLOS ONE

Dear Dr. Ye,

Thank you for submitting your manuscript to PLOS ONE. After careful consideration, we feel that it has merit but does not fully meet PLOS ONE’s publication criteria as it currently stands. Therefore, we invite you to submit a revised version of the manuscript that addresses the points raised during the review process.

As you will see below, the Reviewer was only partly satisfied with the revised version. The Reviewer points to significant remaining issues about the sample size calculations and EMG standardization procedures between sessions. The Reviewer also insists that the results and discussion should be focused on the physiological (H-reflex) rather than subjective measures. After examining the manuscript, I have to agree with the Reviewer on most points. I am particularly concerned by the statistical approach, which seems inappropriate. Why a 'repeated measures ANOVA' for post-test, where you should have used a Tukey's unless you used a multivariate approach. In the latter case, you need to describe it clearly in the text. Can you provide a clear description of which variables (Hmax, Mmax, H/M ratio, VAS) were entered in the ANOVA and whether assumptions were met (Levene's test)? Finally, I am concerned by the values of Mmax reported in your manuscript (about 2.5 mV, p.14v, line 314), which are much lower than those reported Mmax for the Soleus (> 10 mV). Providing examples of H and M recording traces between conditions could help resolve the issues. In the next round, please make sure that all concerns, problems and suggestions are fully addressed.

We look forward to receiving your revised manuscript.

Kind regards,

François Tremblay, PhD

Academic Editor

PLOS ONE

Reviewers' comments:

Reviewer's Responses to Questions

**Comments to the Author**

1. If the authors have adequately addressed your comments raised in a previous round of review and you feel that this manuscript is now acceptable for publication, you may indicate that here to bypass the “Comments to the Author” section, enter your conflict of interest statement in the “Confidential to Editor” section, and submit your "Accept" recommendation.

Reviewer #1: (No Response)

2. Is the manuscript technically sound, and do the data support the conclusions?

Reviewer #1: Yes

3. Has the statistical analysis been performed appropriately and rigorously? 

Reviewer #1: Yes

4. Have the authors made all data underlying the findings in their manuscript fully available?

Reviewer #1: Yes

5. Is the manuscript presented in an intelligible fashion and written in standard English?

Reviewer #1: Yes

6. Review Comments to the Author

Reviewer #1: Dear authors, the current version of the manuscript was improved. However, I still have several comments that might help to improve the quality of you paper. Please consider my suggestions presented below.

Specific comments:

INTRODUCTION

• Line 68: instead of ‘preferred length’, perhaps ‘a specific length that might represent a desired tape tension’ would be more adequate here.

• Line 69: please change ‘may’ to ‘allegedly’, as the mechanisms of KT are not fully proved by research.

• Line 74: please erase ‘muscle fascia and somatic elements’ and change to ‘neuronal excitability due to a tactile stimulus’ (this is the premise of KT).

• Line 74-75: the use of a term such as ‘gold standard’ is mistaken. That would imply an established validity and reliability, which is not the case. At most, you should use something like a ‘lack of standardization’ regarding tape tension.

• Line 125: change ‘can’ to ‘might’.

• Lines 127-129: I suggest removing this last sentence.

• Line 136: erase the term ‘gold standard’, as this is not adequate nor applies to this context. Nevertheless, I suggest removing the last sentences (lines 135-138).

METHOD

• Thanks for the inclusion of the familiarization procedure. However, please be more specific on how and what was adopted as familiarization?

• The sample size calculation description improved; however, it is not clear why you used ‘gait velocity’ as your outcome of interest? This is not adequate, as you should include a suitable variable, for instance, measures of h-reflex based on previous studies. Also, your design would suggest a F test (ANOVA), and not a t test (two dependent means), as you have 3 repeated measures (0, 50 and 100KT).

• Please include more information regarding the EMG acquisition: common mode rejection ration, signal to noise ratio, input impedance, and electrode descriptions (if active or not, and distance between poles). Also, what were the procedures adopted before coupling the electrodes, to improve the signal acquisition and to reduce skin impedance? How did you guarantee the reliability of the EMG electrode placement, considering the different visits to the laboratory? Please clarify.

• Line 237: what do you mean by “facilitated KT application”? Do you mean that the application followed the facilitation technique proposed by the KT method? This is not clear. Also, what process can be used to avoid bias? This information is lost.

• It still not clear how did you determine the 100KT tension without “fracturing” the tape and compromising its elastic properties. This “pre-experiment” to determine the tension should be detailed, at least within a supplement of the paper. If you applied a length-tension curve and determined the maximum tension, the so-called 100% would be the fracturing point? Hence, the 100KT is not actually the maximum tension, whereas it would make more sense to use the greatest tension before reaching the fracture point (you could call that 100KT for the purpose of your experiment, but this information needs clarification). In addition, what was the reliability within this “pre-experiment”?

• I still do not agree with the inclusion of the VAS measurement. From my standpoint, this is a subjective measurement and the focus of your study is the h-reflex measurement. By including the VAS you are dealing with another topic that was not covered by your study rationale and does not add to your research question. My recommendation is to erase this data, as this information does not add to the clinical practice, instead, this might bias the reader as they might wrongly assume that a “feeling” of tension could represent an effect. This is not the case and you should refrain from subjective measurements at this point. The h-reflex might be of more value if you successfully “translate” the clinical and practical findings provided by your study.

RESULTS

• Overall, the results section improved.

• In table 1, perhaps my suggestion was not clear in the previous review: please include the “raw” data (mean and SD), for each variable on each condition (0KT, 50KT and 100KT) stratified by lying and standing. Besides the “raw” data, then include the mean difference together with the CI95% for each condition. This is an adequate presentation of your results, which my help the reader and other researchers to interpret your findings. Moreover, by including this detail on Table 1, I suggest removing Fig 4.

DISCUSSION

• Lines 364-378: As I have suggested, please remove the VAS findings and this discussion, which does not add to your research question and might bias the reader. I suggest focusing on neuronal excitability and posture variations.

• Line 393: The other studies included in the introduction should also be considered in the discussion. This might put your data into context by comparing the studies with your data. Interestingly, even though the tensions were different between studies, most of then have similar findings compared to your study (KT does not influence motoneuronal excitability). Could you discuss further?

• I would suggest a more deepened discussion on why the tape tension did not influence motoneuronal excitability. What might explain your findings? What are the clinical interpretations of your findings?

• Please remove the mention to subjective measurements. The discussion mentions quite frequently what the subjects felt, and this should be improved to a more deepened discussion on physiological aspects and underlying assumptions adopted by the KT method, which was not confirmed by your data.

7. PLOS authors have the option to publish the peer review history of their article (what does this mean?). If published, this will include your full peer review and any attached files.

Reviewer #1: No

---

## [Author Response · Author response to Decision Letter 1]

20 Jun 2020

Please see our Responses to Reviewers file

---

## [Decision Letter · Decision Letter 2]

7 Jul 2020

PONE-D-20-05949R2

Acute Effects of Kinesiology Tape Tension on Soleus Muscle H-reflex Modulations during Lying and Standing Postures

PLOS ONE

Dear Dr. Ye,

Thank you for submitting your manuscript to PLOS ONE. After careful consideration, we feel that it has merit but does not fully meet PLOS ONE’s publication criteria as it currently stands. Therefore, we invite you to submit a revised version of the manuscript that addresses the points raised during the review process.

As you will see, the Reviewer was mostly satisfied with the latest revisions. I concur with the Reviewer that the current version is much improved over the previous one. The Reviewer has only a few minor revisions that should be easily implemented. I advise you to proceed with diligence and re-submit a revised version so a final decision can be made regarding your manuscript. 

We look forward to receiving your revised manuscript.

Kind regards,

François Tremblay, PhD

Academic Editor

PLOS ONE

Reviewers' comments:

Reviewer's Responses to Questions

**Comments to the Author**

1. If the authors have adequately addressed your comments raised in a previous round of review and you feel that this manuscript is now acceptable for publication, you may indicate that here to bypass the “Comments to the Author” section, enter your conflict of interest statement in the “Confidential to Editor” section, and submit your "Accept" recommendation.

Reviewer #1: All comments have been addressed

2. Is the manuscript technically sound, and do the data support the conclusions?

Reviewer #1: Yes

3. Has the statistical analysis been performed appropriately and rigorously? 

Reviewer #1: Yes

4. Have the authors made all data underlying the findings in their manuscript fully available?

Reviewer #1: Yes

5. Is the manuscript presented in an intelligible fashion and written in standard English?

Reviewer #1: Yes

6. Review Comments to the Author

Reviewer #1: General comments:

The current version of the manuscript was improved and it is suitable for publication. I have few minor suggestions at this time.

Specific comments:

DISCUSSION

• Lines 423-425: please remove the sentence “Therefore, for clinicians such as physical therapists…”. This sentence does not add, as the previous one is objective enough and shows that tape tension does not provide meaningful effects to motoneuron excitability.

• Lines 459-460: The sentence “KT application altered tension perception” should be removed, considering that the VAS data was removed from the manuscript. Please revise the text accordingly and remove any mention to the subjective measurements.

• Lines 466-468: please remove the sentence: “Thus, when applying the KT to individuals, the tension to be used should not be a concern in terms of modulating the sensorimotor activity ascribed to elastic taping”.

7. PLOS authors have the option to publish the peer review history of their article (what does this mean?). If published, this will include your full peer review and any attached files.

Reviewer #1: No

---

## [Author Response · Author response to Decision Letter 2]

7 Jul 2020

The responses to comments file has been uploaded with the revised manuscript.

---

## [Editor Report · Decision Letter 3]

10 Jul 2020

Acute Effects of Kinesiology Tape Tension on Soleus Muscle H-reflex Modulations during Lying and Standing Postures

PONE-D-20-05949R3

Dear Dr. Ye,

We’re pleased to inform you that your manuscript has been judged scientifically suitable for publication and will be formally accepted for publication once it meets all outstanding technical requirements.

Kind regards,

François Tremblay, PhD

Academic Editor

PLOS ONE
---

## [Editor Report · Acceptance letter]

14 Jul 2020

PONE-D-20-05949R3 

Acute Effects of Kinesiology Tape Tension on Soleus Muscle H-reflex Modulations during Lying and Standing Postures 

Dear Dr. Ye:

I'm pleased to inform you that your manuscript has been deemed suitable for publication in PLOS ONE. Congratulations! Your manuscript is now with our production department. 

Kind regards, 

on behalf of

Dr. François Tremblay 

Academic Editor

PLOS ONE